# A Robust Framework Combining Image Processing and Deep Learning Hybrid Model to Classify Cardiovascular Diseases Using a Limited Number of Paper-Based Complex ECG Images

**DOI:** 10.3390/biomedicines10112835

**Published:** 2022-11-07

**Authors:** Kaniz Fatema, Sidratul Montaha, Md. Awlad Hossen Rony, Sami Azam, Md. Zahid Hasan, Mirjam Jonkman

**Affiliations:** 1Health Informatics Research Lab, Department of Computer Science and Engineering, Daffodil International University, Dhaka 1207, Bangladesh; 2College of Engineering, IT and Environment, Charles Darwin University, Darwin, NT 0909, Australia

**Keywords:** ECG images, cardiovascular disease, image preprocessing, transfer learning models, deep convolutional neural network, ablation studies, k-fold cross validation

## Abstract

Heart disease can be life-threatening if not detected and treated at an early stage. The electrocardiogram (ECG) plays a vital role in classifying cardiovascular diseases, and often physicians and medical researchers examine paper-based ECG images for cardiac diagnosis. An automated heart disease prediction system might help to classify heart diseases accurately at an early stage. This study aims to classify cardiac diseases into five classes with paper-based ECG images using a deep learning approach with the highest possible accuracy and the lowest possible time complexity. This research consists of two approaches. In the first approach, five deep learning models, InceptionV3, ResNet50, MobileNetV2, VGG19, and DenseNet201, are employed. In the second approach, an integrated deep learning model (InRes-106) is introduced, combining InceptionV3 and ResNet50. This model is developed as a deep convolutional neural network capable of extracting hidden and high-level features from images. An ablation study is conducted on the proposed model altering several components and hyperparameters, improving the performance even further. Before training the model, several image pre-processing techniques are employed to remove artifacts and enhance the image quality. Our proposed hybrid InRes-106 model performed best with a testing accuracy of 98.34%. The InceptionV3 model acquired a testing accuracy of 90.56%, the ResNet50 89.63%, the DenseNet201 88.94%, the VGG19 87.87%, and the MobileNetV2 achieved 80.56% testing accuracy. The model is trained with a k-fold cross-validation technique with different k values to evaluate the robustness further. Although the dataset contains a limited number of complex ECG images, our proposed approach, based on various image pre-processing techniques, model fine-tuning, and ablation studies, can effectively diagnose cardiac diseases.

## 1. Introduction

Cardiovascular disease (CVD) can be a silent disease [1] that poses a severe threat to human health, especially for older and middle-aged people [2,3,4,5]. High cholesterol, increased triglyceride levels, high blood pressure, obesity, etc., increase the risk of heart disease [6]. In 2015, approximately 20 million people died due to heart disease [7]. It is estimated that approximately 23.6 million people will die of CVD every year by 2030 [5]. It is apprised that in the UK, 7.6 million people live with various cardiac diseases; 3.6 million females and 4 million males; moreover, approximately 47,000 people under the age of 75 die of heart diseases [8]. It was recorded that patients affected by COVID-19 can suffer from various cardiac diseases that may lessen their long-term chance of survival [9]. Early detection of heart disease is crucial because early identification and timely treatment play a vital role in increasing the chances of survival [10]. Electrocardiography is the most common technology to identify heart disease [11]. A visual study of paper-based electrocardiogram (ECG) waveforms by specialists is still one of the most prevalent techniques. However, analyzing such massive volumes of data is time-consuming and error-prone. Moreover, in many rural areas around the world, a lack of expert clinicians is an obstacle to timely diagnosis. A large number of ECG records are still available in the paper form [12], and physicians and medical researchers examine paper-based ECG images to diagnose cardiac diseases [13]. An automated, reliable, accurate, and quick diagnostic method using paper-based ECG images could aid clinicians and may reduce mortality. We propose an automated method to classify paper-based electrocardiogram signals (ECGs) using a deep learning model [14].

This study uses a dataset of paper-based ECG images of cardiac and COVID-19 patients [13] and aims to classify these into myocardial infarction (MI), history of myocardial infarction (HMI), normal heartbeat (NHB), abnormal heartbeat (AHB), and COVID-19. Though there are a limited number of images in this dataset, we did not apply any augmentation techniques. Different image pre-processing techniques, cropping, morphological opening, contour detection, Gaussian blur, and non-local means (NLM) algorithms are applied to the raw images to remove the artifacts and background graph lines of the ECG images. The histogram equalization (HE) enhancement technique is applied to enhance image quality.

After pre-processing the dataset, two approaches are taken in order to generate a model. In the first approach, five deep learning models are employed. In the second approach, a hybrid deep convolutional neural network (DCNN) architecture is proposed so that the model can extract and learn more deep and hidden features and classify accordingly. The research aim, major contributions, and working processes in this study are described below.

## 2. Research Aims and Major Contribution of the Paper

Several researchers have applied existing deep learning models such as InceptionV3, ResNet50, MobileNetV2, VGG19, and DenseNet201 to classify medical images, including ECGs. However, there are limitations in terms of multi-class classification, particularly of paper-based ECG images. Since paper-based ECG images usually contain various artifacts and low-contrast images of uneven size, digital ECG signals are easier to work with than paper-based images. Though several studies [15,16,17,18] utilized paper-based image datasets, there were some major limitations. For example, in some studies, researchers did not use image preprocessing techniques although the dataset contained various artifacts and low-contrast images. They also created a completely balanced dataset, although the interpretation capability of a model might not be evaluated efficiently employing a completely balanced dataset. These limitations might affect classification accuracy and performance. These limitations are addressed in this study. We have used a paper-based ECG image dataset that is highly imbalanced and contains images with different resolutions, low contrast, and noise. Working with a highly imbalanced dataset without using augmentation techniques is challenging, but augmentation techniques can degrade the model’s accuracy to classify cardiac diseases [17]. We eliminated noises from the images, enhanced the brightness and contrast of low-resolution images, and proposed a robust model framework that was able to perform well on this imbalanced dataset. After employing the InceptionV3, ResNet50, MobileNetV2, VGG19, and DenseNet201 models, we integrated the two best convolutional neural network (CNN) models (InceptionV3 and ResNet50), based on the highest accuracy of the five CNN models, to form a base DCNN model. An ablation study was performed on the base DCNN model to develop a robust model framework with an optimal configuration that can accurately detect cardiac diseases. It can be inferred that using a well-pre-processed ECG image dataset by applying proper image pre-processing techniques and creating a robust DCNN model by performing an ablation study are crucial steps of this research.

The main contributions of this paper can be summarized as follows:As paper-based ECG images often contain artifacts, including unnecessary text considered labels and background graph lines, that may interfere with the model’s performance, an automated cropping system and different image preprocessing techniques are employed to remove these labels and graph lines, respectively. Furthermore, the contrasts and brightness are balanced by applying HE image enhancement techniques.Quantitative evaluations, such as peak signal-to-noise ratio (PSNR), mean squared error (MSE), root mean squared error (RMSE), and structural similarity index measure (SSIM), are utilized with pre-processed cardiac images to ensure that image quality is not reduced.An integrated deep learning model (InRes-106) is proposed, combining InceptionV3 and ResNet50 to improve the performance compared to existing models.An ablation study of ten fields is implemented to further develop the proposed model by changing the model architecture and hyperparameters while taking time complexity into account without compromising accuracy.Performance metrics, such as sensitivity, specificity, precision, recall, accuracy, F1-score, false positive rate (FPR), false negative rate (FNR), false discovery rate (FDR), mean absolute error (MAE), RMSE, Matthews correlation coefficient (MCC), kappa coefficient (KC), a Wilcoxon signed-rank test, and receiver operating characteristic curve (ROC), are calculated to evaluate the results.To evaluate the robustness of our proposed network, InRes-106, the model is trained using the k-fold cross-validation method with k values of 1, 3, 5, 7, and 9.

## 3. The Literature Review

The early identification of cardiac disease and classification of ECG images with the help of various deep learning classifiers is currently a major area of research. Several researchers have proposed various deep learning methods to classify cardiac diseases using ECG images or ECG signals. We reviewed some studies [15,16,17,18] that applied different deep learning techniques to paper-based ECG images and some studies [19,20,21,22,23,24] that used digital ECG signals to classify cardiac diseases. Mehmet et al. [15] used the cardiac and COVID-19 paper-based ECG image dataset to classify COVID-19 ECGs. They proposed a modified AlexNet model and divided their work into two scenarios. In the first scenario, they performed a binary classification of COVID-19 and normal heart beat (NHB). For binary classification, the model performed well and identified COVID-19 with an accuracy of 96.20%. In the second scenario, they labeled the NHB, AHB, and MI classes as negative and the COVID-19 class as positive, resulting in a 93.00% classification accuracy. A similar study was performed by Irmak [16], where a DCNN model was proposed to perform binary and multi-class classification using ECG trace images. They achieved the highest accuracy of 98.57% for COVID-19 and normal, 93.20% for COVID-19 and AHB, and 96.74% for COVID-19 and HMI in several binary classifications. However, in multi-class classification, an 86.55% accuracy was achieved for COVID-19, AHB, and HMI and an 83.05% accuracy was achieved for normal, COVID-19, AHB, and HMI. In another study [17], a deep learning model was applied to the same paper-based ECG image dataset for identifying COVID-19 and other heart diseases. They compared testing accuracy before and after the application of multiple augmentation techniques in their work and observed that the accuracy dropped from 81.8% before augmentation to 76.4% after augmentation. They concluded that augmentation techniques do not play a major role in predicting cardiac diseases. Several studies [15,16,17] created a balanced dataset by modifying the number of images. They addressed binary and multi-class classification but the main focus was on identifying COVID-19. However, while their proposed model achieved promising results in binary classification, the accuracy was not good enough in multi-class classification. Khan et al. [18] proposed a cardiac disorder detection system from 12 lead paper-based ECG images. A single shot detector (SSD) MobileNetV2 architecture was used to classify the images into four classes: normal, AHB, MI, and HMI. The proposed model achieved an accuracy of 98%. They identified only four classes while they converted their 929 real images into 11,184 images, using a segmentation technique corresponding to the 12 leads of the ECG. Several other studies used digital ECG signals for disease classification. Wahyu et al. [19] proposed a method to detect and predict diseases for real-time ECG signals using the PhysioNet open-source database. The authors first used a CNN model to classify and predict four different classes: normal, sudden death, arrhythmia, and supraventricular arrhythmia. Their model achieved 95% accuracy on 100 epochs. Mahwish et al. [20] proposed a deep learning method for identifying ventricular arrhythmia (VS) and by converting the ECG signals into images. They first transformed the ECG signals into 32 × 32 binary images, then obtained 126,976 values after normalization of the images and applied AlexNet, Inception-v3, and VGG16 models for training purposes. Afterwards, the features were merged by applying a DCNN model, and the optimal features were selected utilizing a heuristic entropy calculation method. Supervised learning models were used for the final feature selection. The Massachusetts Institute of Technology-Beth Israel Hospital (MIT-BIH) dataset was used in this recent work, and their proposed model achieved an accuracy of 97.6%, a sensitivity of 98.2%, a specificity of 97.5%, and a F-score of 0.979 while using the cubic support vector machine as a final stage classifier. Jinyong Cheng et al. [21] developed a dense heart rhythm model that integrates a 24 layer DCNN and bidirectional long short-term memory (BiaLSTM) to find the hierarchical and time-sensitive features of ECG images. Three different convolution kernels were applied to find the detailed ECG features. A combination of wavelet transforms and median filtering was used to remove noise from the original ECG signals. Applying a ten-fold cross validation, their proposed model achieved an accuracy of 89.3% and a F1-score of 89.1%. In another study [22], researchers combined generative adversarial network (GAN) and long short-term memory (LSTM) and developed an ensemble model that was applied and tested to two different datasets (MIT-BIH and PTB-ECG) of ECG signals. For the MIT-BIH dataset, their model achieved an accuracy of 99.4% and an F1-score of 98.7%. For the PIT-ECG dataset, their system achieved an accuracy of 99.4% and an F1-score of 99.3%. Nahian et al. [23] proposed two approaches, namely, empirical modified decomposition (EMD) and higher-order intrinsic mode functions (IMFs), to decompose the ECG signals. They used three public ECG datasets: MIT-BIH, PTB, and St Petersburg. The model achieved an accuracy of 97.70% for the MIT-BIH dataset, 99.71% for St Petersburg, and 98.24% for the PTB dataset. It can be concluded that most of the studies presented above did not explore image pre-processing techniques extensively and that experimenting with different hyperparameters was also missing. The models may be further improved by performing an ablation study and exploring different image pre-processing methods.

## 4. Methods

Artifact removal and enhancing image quality are crucial in order to acquire optimal performance from a deep learning model. We, therefore, applied several image processing techniques to remove artifacts and enhance quality before training the model. The flow diagram of cardiac disease classification, including image pre-processing and model building, is shown in Figure 1. All processes are explained in more detail in the following sections and sub-sections.

The cardiac image dataset contains artifacts, such as unnecessary text information, on the ECG records. To remove these, an automated cropping method was employed. Different image pre-processing techniques were applied to remove background graph lines on the ECG images. To balance the brightness and contrast level, the HE enhancement technique was applied. After pre-processing the dataset, two approaches were introduced. In the first approach, five deep learning models were employed, and the performance was recorded. In the second approach, a hybrid DCNN architecture, named InRes-106 combining InceptionV3 and ResNet50, was proposed, followed by an ablation study.

### 4.1. Dataset

A total of 1932 paper-based ECG images comprising five classes named MI, HMI, NHB, AHB, and COVID-19 were analyzed for this study. This is a publicly available dataset [13]. A total number of 12 lead systems were visible in each ECG report. The dataset contains signals that were collected using the EDAN SE-3 series 3-channel electrocardiograph [15]. The dataset contains 546 images of AHB patients, 250 images of COVID-19 patients, 203 images of HMI patients, 74 images of MI patients, and 859 images of NHB patients. Figure 2 illustrates the ECG images of five classes.

The paper-based ECG images consist of 12 leads: I, II, III, aVF, aVR, aVL, V1, V2, V3, V4, V5, and V6. The ECG signal consists of waves, the P wave, the QRS complex, and the T wave. However, in this dataset, lines (graph line) are also attached to the background and some labels are found on the ECG reports. Moreover, there are a number of low contrast signal images in the dataset. Figure 3 illustrates some challenges of the dataset, including artifacts, noise, and low contrast images (challenges are marked in red).

### 4.2. Image Pre-Processing

Image pre-processing is considered an important task before feeding the images to neural networks. Without image pre-processing methods, an acceptable classification performance might not be achievable for paper-based ECG images, as interpreting ECG images is challenging due to the complex pattern of signal and the presence of artifacts and noise. ECG images contain various artifacts and low-contrast images. Training our deep learning model without any image preprocessing images may interfere with the performance. In contrast, an optimal image preprocessing system can easily solve this problem by performing exceptionally well. Therefore, the main motivation for these steps is to remove artifacts and improve image quality in our study. In this section, all the pre-processing steps are described in sequence, including artifact removal and image enhancement. Figure 4 illustrates the main processes and sub-processes of the ECG image pre-processing stage. The output of the previous step is the input for the next step (e.g., the output of the cropped images is the input of the first operation of graph line removal).

The artifacts (labels and background graph lines) that are present in the ECG image are removed by applying cropping, morphological opening, contour detection, Gaussian blur, and NLM. Afterwards, the HE enhancement technique is applied to the artifact removed dataset to enhance the image quality [24].

#### 4.2.1. Artifacts Removal

The ECG images contain unwanted areas, in particular, labels and graph lines. Therefore, this phase is divided into two sub-phases: label removal and graph line removal.

##### Removal of Labels

Generally, in an ECG strip, some labels are found on the upper and the bottom portion of images. They are considered artifacts since they may interfere with the classification of the images. Therefore, the images are cropped, using an automated cropping system, to remove these artifacts. As a result, the pixel size for all the images is reduced from the original size of 2213 × 1572 to 2058 × 1210. The value of the cropping ratio is chosen in such a way that the artifacts are removed while preserving the necessary information. Figure 5 shows the label-removal process following the masking image and contouring image methods sequentially. It can be seen that unwanted labels are successfully removed in the contour image output while preserving the region of interest (ROI). A similar cropping system can be applied to other datasets with unwanted text or labels.

##### Graph Line Removal

After removing the labels using the cropping method, Otsu thresholding, morphological opening, contour detection, inverse contour, Gaussian blur, and NLM are applied to remove the background graph lines from the images. In this case, we applied the trial-and-error method to select all the optimal parameters that would perform best for our dataset. Moreover, finetuning is an important aspect to obtain the best parameters for producing the best results. After reading an ECG image, it is converted to the grayscale format. Otsu thresholding is applied to the grayscale image to convert it to binary format as morphological opening provides optimal output when applied to a binary image. Morphological opening is applied to the binary image to remove noise. A rectangular kernel with a size of (2, 2) is used. As kernel structure can be varied according to the characteristics of the images [25], the kernel shape and size were determined after experimenting with different values. Afterwards, all contours are detected from the output image of morphological opening using the findContours method. Then, only the contours with pixel size < 50 are drawn using drawContours function so that only the graph lines are drawn while ignoring the ECG signals. Finally, before applying Gaussian blur, the output image after drawing the contours is inversed by subtracting the pixels values from 255. We applied Gaussian blur on the inversed image with a pixel parameter value of (3, 3) to blur the background lines and remove noise from images [26,27]. After that, NLM is applied to the blurred images to remove the blurry background lines from the image; moreover, this technique is used to provide an improvement of the signal to noise ratio [28]. This algorithm is applied using two functions, namely, estimate_sigma and denoise_nl_means. The estimate_sigma function is used to provide an optimal starting point to set the parameters for NLM algorithms. After that, the noisy ECG signals are denoised by applying the denoise_nl_means function after experimenting with different parameter values [29]. In this regard, the parameter values help to control the weight loss of the patch as a distance function between different patches in images, where larger parameter values allow for smoothing between dissimilar patches. Figure 6 shows the graph line removal process and the intermediate outcomes of each stage.

#### 4.2.2. Image Enhancement

After removing artifacts from the images, the HE technique is applied to the pre-processed dataset to adjust brightness and contrast [30]. In this process, the lost contrast of an image can be restored by remapping the brightness values [31]. It can help to enrich the contrast of the images and smooth the pixels. The outcome of HE is shown in Figure 7.

The process of HE is as follows [32]:
Calculate the pixel number in each grey level of the real input image. The jth value represents the grey level that is given by ni. Here, j = 0, 1,…, N−1, and N denotes the total number of grey levels.Compute the histogram of the real image, presented as the possible density of each grayscale given $Q_{j}\;\left(b_{j}\right)=m_{j}/m,$ where m indicates the total number of pixels in the actual image.Estimate the cumulative distribution function as follows: $t_{k}b_{k}\approx \;\sum_{\left(j=0\right)}^{k}Q_{j}\left(b_{j}\right),k=0,1,\ldots \;,N-1$.Compute the final result grayscale $h_{k}=\frac{INT\left[\left(h_{max}-h_{min}\right)t_{k}\left(b_{k}\right)+h_{min}+0.5\right]}{N-1},\;k=0,1,\ldots \;,N-1$ and $INT\left[.\right]$, which is an integer operator. Let $h_{min}=0$ and $h_{max}=N-1$. This Equation can be simplified into $h_{k}=\frac{INT\left[\left(N-1\right)t_{k}\left(b_{k}\right)+h_{min}+0.5\right]}{\left(\left(N-1\right)\times \left(I\right)\right)}$.As per the mapping relation between $e_{k}$ and the real image grayscale function and $h_{k}$, alter the real grayscale to obtain an output image in which the histogram is nearly equally distributed.

#### 4.2.3. Assurance of Image Quality

Due to applying the algorithms described above, the image quality could be reduced in some cases. We, therefore, calculate the MSE, SSIM, PSNR, and RMSE values, comparing the source images and the pre-processed images to ensure that the image quality is not reduced. Generally, the MSE range is between 0 to 1, where a value greater than 0.5 indicates that the images are of good quality, and a value less than 0.5 indicates a low-quality image. The SSIM score ranges from −1 to 1. A value of 1 indicates an exact structural match, and −1 indicates the opposite [33]. The optimal PSNR ranges from 30 to 50 dB for 8-bits images. A value below 20 dB is unsatisfactory [34]. RMSE calculates the qualitative difference between real and processed images and a lower RMSE value, especially around 0, means less error and better image quality [25]. Figure 8 illustrates the difference between the original image (after label removal) and the final enhanced image (after preprocessing).

We used the original and enhanced images of the same signal to calculate MSE, PSNR, SSIM, and RMSE for a total of 1932 images, but it is not convenient to present all the values. We, therefore, randomly select twenty images for an indication of the values, as seen in Table 1. For all 1932 images, the average value of MSE is 0.147, PSNR is 36.98, SSIM is 0.945, and the RMSE is 0.123. A pie chart is also presented in Figure 9 to visualize the percentage of the 1932 images according to the range of PSNR values. The values of MSE, PSNR, SSIM, and RMSE for twenty randomly images are shown in Table 1.

As can be observed, the MSE and RMSE values are close to 0, PSNR values are above 36, and SSIM values are close to 1.

Figure 9 shows that approximately 41.60% of images; which is a large proportion, have PSNR values between 36.01, and 37.00. 35.10% of images have PSNR values between 37.01 and 38.00, 10.96% of images between 35.01 and 36.00, 9.32% images between 38.01 and 39.00, and 3.02% images between 39.01 and 40.00. It can be inferred from this pie chart that most images are within an acceptable range. The MSE and RMSE are close to 0, and the SSIM value is close to 1. This demonstrates that the image quality is well preserved, even after applying the pre-processing algorithms.

### 4.3. Proposed Approaches

Two approaches are evaluated. In the first approach, five deep learning models, InceptionV3, ResNet50, MobileNetV2, VGG19, and DenseNet201, are applied. In the second approach, we combine the models (InceptionV3 and Reset50) as an integrated deep learning model intending to improve the performance. Before feeding the images to the model, the dataset is split.

#### 4.3.1. Dataset Split

After pre-processing the ECG images, the dataset is split into the training, validation, and testing sets, using a ratio of 70:20:10, respectively. The models were evaluated with 1351 images as a training set, 384 images as a validation set, and 197 images as a test set. The splitting ratio is kept the same for both approaches. A GitHub link with the IDs of the selected ECG images for training, testing, and validation in this study is shared here [35].

#### 4.3.2. First Approach

In this approach, five pre-trained models, named InceptionV3, ResNet50, MobileNetV2, VGG19, and DenseNet201, are trained and evaluated to observe their performance in terms of accuracy.

##### ResNet50

ResNet50 consists of 50 layers (48 convolutional layers, 1 average pool layer, and 1 max pool layer) that contain approximately 23 million parameters [36]. The model is widely used in computer vision problems due to the core advantage of the architecture, which helps to reduce the vanishing gradient problems by following the alternate shortcut path. The model performs basic functions such as convolution and max-pooling by following stacked convolutions. This stacked convolution provides an advantage in solving the gradient vanishing problems [37].

##### InceptionV3

This pre-trained model contains 48 layers and 23 million trainable parameters [38]. It consists of convolutional, max-pooling, average pooling, concats, dropout, and fully connected (FC) layers. The pointwise 1 × 1 convolution is used in this architecture to reduce the number of parameters and the computational cost. The model helps to reduce the number of parameters by factoring in the convolution without reducing the efficiency of the performance features [36].

##### MobileNetV2

MobileNetV2 architecture consists of 53 layers with 3.5 million trainable parameters [39]. It consists of two types of blocks, and every single block has three layers. The first and third layers in both blocks are 1 × 1 convolutional layers with 32 filters, and the middle (second) layer is a depth-based convolutional layer. The longitudinal bottlenecks between the layers play an essential role to prevent nonlinearity from massive data loss [25].

##### DenseNet201

This model architecture consists of 201 convolutional layers, 98 route layers, one average pooling layer, four max-pool layers, and Softmax [40]. It contains a total of 6,992,806 trainable parameters [41]. DenseNet201 is widely used because of several advantages such as strong gradient flow, computational efficiency, and diverse features. Moreover, the model significantly reduces the number of parameters and the vanishing-gradient problem [42].

##### VGG19

VGG19 is a variant of the VGG model with a total of 19 layers and 50,178 trainable parameters [43]. It has three additional FC layers at the end of the VGG16 model consisting of 4096, 4096, and 1000 neurons, respectively, making it a total of 19 layers. Additionally, the convolutional layers are arranged with the rectified linear unit (ReLU) activation function [25].

##### Training Approach

To train the models, the maximum number of epochs is set to 160 and the batch size to 32. The Adam optimizer is utilized with a learning rate of 0.001. Categorial cross-entropy is used as a loss function. As our paper-based ECG image dataset contains five classes, we have changed the number of classes to five in the FC layer.

#### 4.3.3. Second Approach

In order to achieve a better performance, we introduce an integrated deep learning model (InRes-106) by combining InceptionV3 and ResNet50 as a DCNN model because it can be assumed that the fine-tuning InceptionV3 and ResNet50 model architectures respond well to the medical images [44]. The advantage of DCNN is that a high accuracy can be achieved through multiple levels and an automated feature extraction process [45]. In addition, it is able to extract hidden and high-level features from the image. It recognizes images based on shapes as well as complex structures and features.

After building a base model, an ablation study is performed in order to achieve the optimal model configuration, based on performance. This is achieved by changing different components, such as the activation function, hyperparameters, the loss function, and the flatten layer. The training approach is the same as for the transfer learning models. To train the model, the maximum number of epochs is set to 160, the batch size to 32, ReLU is used as the activation function, and categorial cross-entropy as the loss function. The Adam optimizer is utilized with a learning rate of 0.001.

##### Base Model

The base model consists of a total number of four blocks, one average polling, and one FC layer. The input image size is fixed at 224 × 224 for this model. The initial block (Block-1) contains three convolution layers with 32 channels of 2 × 2 kernel size followed by a max-pooling layer. Likewise, the two convolution layers have 32 channels of 2 × 2 kernel, again followed by a max-pooling layer. The next block (Block-2) contains a combination of the InceptionV3, and ResNet50 blocks, whereas the inception block contains six convolution layers with 32 channels of 2 × 2 kernel size, followed by one max-pooling layer, and the ResNet block consists of only three convolutional layers. This block is stacked three times (3 × Inception-ResNet Block A), so that a total number of 30 layers is present in Block-2. The next block (Block-3) contains five convolutional layers, followed by a max-pooling layer in the inception block, and the ResNet block again consists of only three convolutional layers. This block (Block-3) is also stacked five times, so a total number of 45 layers are generated. Likewise, the last block (Block-4) consists of four convolutional layers with 32 channels of 2 × 2 kernel size, followed by a max-pooling layer in the inception block and three convolution layers for the ResNet block. This block (3 × Inception-ResNet Block C) is also stacked three times; resulting in 28 layers. Each block is created in a sequential process so that it contains a feature map, an inception block, a ResNet block, a ReLU activation function, and one more feature map. After creating all the blocks, one average pooling layer and one FC layer are added to the last block, which outputs five channels for the five classes. The ReLU function is applied as the activation function and average pooling as the flatten layer. Figure 10 illustrates the base model of our study.

##### Ablation Study

Although the existing CNN models are well established, it is not certain that all CNN models perform well on any given dataset. On the other hand, the model’s performance can be strengthened by changing the hyperparameters. In general, an ablation study is performed to test the stability and functionality of a CNN model by altering and removing different layers and hyperparameters. When a change is introduced into the model architecture, the network can show increased, decreased, or identical performance. Often, the accuracy can be boosted by examining various hyperparameters such as learning rates, loss functions, batch size, and optimizers. Changes in the model architecture to address the computational complexity issue can also affect the overall performance. A possible reduction in the model’s performance may be fixed by updating and fine-tuning the model. Our main concern was to build a robust DCNN model by changing all hyperparameters through an ablation study method and, where possible, to reduce the computational complexity without compromising the accuracy. In this regard, ten ablation study cases are applied to the base model architecture to achieve the optimal configuration of the proposed DCNN model by changing the filter size, the number of filters, the pooling layer, the activation function, the batch size, the flatten layer, the loss functions, the optimizers, and the learning rate, while reducing computational complexity. This is a sequential process. Initially, we set a standard hyperparameter value for the first study and proceeded with the experimentation by changing values. After obtaining the optimal hyperparameter from the first study based on the performance, we continued with the second study, keeping the optimal value of the first hyperparameter but changing the hyperparameter for the second study. A similar approach was followed for subsequent studies until we achieved the optimal configurations for our proposed model. All the experimental results are described in Section 5.3.

##### Proposed Model Architecture

Ten ablation studies are performed in order to achieve the optimal model configuration, resulting in the proposed model InRes-106. This is achieved by providing a residual scaling factor (RSF) between InceptionV3 and ResNet50 block, changing the activation function, changing the number of filters and the filter size, changing the average pooling layer, changing the loss function, changing the flatten layer, and changing the batch size, learning rate, and optimizer. Usually, the training set consists of latent information, and in order to train a network so that it is more stable, it is necessary to set the parameter of RSF as trainable as a small value helps to stabilize the training process [46]. Hence, the RSF standard should not be too high as it might impede training. RSF values between 0.1 and 0.3 are selected, see [47]. The initial value is set to 0.1 and is updated during the training process. The optimal RSF value is utilized in the final model. The activation function also has a notable influence on the model performance. The ReLU activation function is commonly used in CNN because of its effectiveness and simplicity. However, the ReLU activation function forces all negative input values to be zero, and as negative gradient values become zero, this can result in loss of information. A parametric rectified linear unit (PReLU) layer may help to overcome these issues [46]. Therefore, we evaluate the results of several activation functions, including the PReLU function. If the input value is negative, the input value will be multiplied by a fixed scaler k. As the scaler value is a hyperparameter, the value needs to be set before training. The PReLU [48] activation functions help to turn the scalar value into a trainable parameter in order to achieve the best performance.

The formula of the function of PReLU is as follows:(1)$$A_{i}=\left\{\begin{array}{c} B_{i}\;,\;\;\;\;\;\;\;\;\;\;\mathrm{if}\;\;B_{i}\;\geq \;0\\ K_{i}B_{i}\;,\;\;\;\;\;\;\;\mathrm{if}\;B_{i}<0 \end{array}\right.$$
where $K_{i}$ defines a learnable parameter scale value, and the remainder are trainable parameters. We initially set the value as 0.1 and let it be trainable so that the value can be updated during the training iterations. Figure 11 illustrates the proposed model architecture.

As in this proposed model, no extra convolutional or max-pooling layer is added to the hybrid architecture; the block number (total 4 blocks) and the layers (total 106) were the same. So, as shown in Figure 11, the number of blocks is the same as in the base model but each block contains one inception block, one RSF trainable parameter, and one ResNet block. Moreover, each block also has its own structure, such as a starting feature map, an inception block, an RSF value, a ResNet block, a ReLU activation function replaced with PReLU, and again, a feature map. In addition, in all the convolution layers from Block 1 to Block 4, the number of filters is set to 64, the average pooling layer is replaced with the flatten layer and then connected to the FC layer. Therefore, each block generates and transfers their output to its next block. Finally, a flatten and FC layer calculate the probabilistic outputs for the five classes to identify and classify the ECGs.

## 5. Results and Analysis

### 5.1. Evaluation Metrics

We applied some metrics to evaluate all the classification models in this study, such as precision, recall, specificity, F1-score, FPR, FNR, FDR, MAE, and RMSE. The confusion matrix for the best model is also shown in Figure 12C. The performance metrics values were calculated using true positive (TP), true negative (TN), false positive (FP), and false negative (FN) values, according to the following equations [49,50]. In the equations for MAE and RMSE, *n* denotes the total number of observations and *y^p^* denotes the predicted value of *y*.
(2)$$recall=\frac{TP}{TP+FN}$$
(3)$$specificity=\frac{TN}{TN+FP}$$
(4)$$precision=\frac{TP}{TP+FP}$$
(5)$$F_{1}\text{-}score=2\frac{precision\times recall}{precision+recall}$$
(6)$$FPR=\frac{FP}{FP+TN}\;\;$$
(7)$$FNR=\frac{FN}{FN+TP}$$
(8)$$FDR=\frac{FP}{FP+TP}$$
(9)$$MAE=\frac{1}{n}\overset{n}{\underset{J=1}{\sum }}\left|y_{j}-y_{j}^{p}\right|$$
(10)$$RMSE=\sqrt{\frac{1}{n}\overset{n}{\underset{J=1}{\sum }}{\left(y_{j}-y_{j}^{p}\right)}^{2}}$$

### 5.2. Results and Analysis of Deep Learning Models

Table 2 presents the performance values (precision, recall, specificity, F1-score, validation accuracy, validation loss, testing accuracy, and testing loss) for five deep learning models with the Adam optimizer and a learning rate of 0.001. As can be seen from Table 2, the InceptionV3 deep learning model recorded the highest testing accuracy of 90.56%, a validation accuracy of 88.45%, a specificity of 90.68%, a precision of 90.86%, a recall of 90.18%, an F1-score of 90.54%, a validation loss of 0.5128, and a testing loss of 0.3209. The ResNet50 model can be considered the second-best because it acquired 89.63% testing accuracy, a validation accuracy of 87.89%, a validation loss of 0.4434, a testing loss of 0.3507, a specificity of 89.19%, a precision of 89.7%, a recall of 89.31%, and an F1-score of 89.32%. DenseNet201 achieved an 88.94% testing accuracy, a validation accuracy of 86.78%, a validation loss of 0.5845, a testing loss of 0.3802, a specificity of 93.59%, a precision of 85.23%, a recall of 81.67%, and an F1-score of 82.90%. VGG19 achieved a testing accuracy, a validation accuracy, a validation loss, a testing loss, a specificity, a precision, a recall, and an F1-score value of 87.87%, 85.36%, 0.7233, 0.3901, 87.63%,81.23%, 79.12%, and 80.37%, respectively. The MobileNetV2 architecture performance was less good than the other models with an 80.56% testing accuracy, a 79.34% validation accuracy, a testing loss of 0.4328, a validation loss of 0.6216, a specificity of 78.94%, a precision of 78.72%, a recall of 77.89%, and an F1-score of 78.23%. After evaluating all the performance metrics, it can be concluded that the InceptionV3 and ResNet50 models performed reasonably well compared to others in detecting cardiac diseases but that the performance was not entirely satisfactory. We, therefore, proceed with our proposed InRes-106 DCNN model.

### 5.3. Results and Analysis of Ablation Study

After training the base hybrid model, we obtained an accuracy of 95.01% (initial accuracy). Our objective was to increase this accuracy while addressing time complexity through a couple of ablation studies. In this section, we show the results of the ablation studies. Table 3 presents the results of altering the RSF value, the filter size, the number of filters, the pooling layer, and the activation function. Table 4 shows the ablation results of altering the batch size, the flatten layer, the loss function, the optimizer, and the learning rate.

Ablation Study 1: Altering Residual Scaling Factor.

We have experimented with our proposed model with and without an RSF value. In the base model, we did not use any RSF value and the testing accuracy reached 95.01% when the training time per epoch was 84 s with a huge time complexity of 657.3 million. When we added an RSF value between InceptionV3 and ResNet50 block, our proposed model achieved an accuracy of 95.59% for a training time per epoch of 77 s and a time complexity of 434.5 million. Therefore, using an RSF value would be beneficial to enhancing the testing accuracy. We, therefore, proceed with this.

Ablation Study 2: Altering Filter size.

We experimented with three different filter sizes of 2 × 2, 3 × 3, and 5 × 5 [51]. Using a 5 × 5 filter size, our proposed model achieved a testing accuracy of 95.894%, with a training time per epoch of 85 s and a time complexity of 717.4 million. Using a 3 × 3 filter size model resulted in a 95.59% testing accuracy, which was very close to the highest accuracy, taking only 77 s per epoch, with a time complexity of 434.5 million. Hence, to achieve less time complexity while maintaining high accuracy, a 3 × 3 filter size is chosen for further ablation studies.

Ablation Study 3: Altering the number of filters.

The kernel number was set to 16, achieving 95.23% testing accuracy while the training time per epoch took 74 s and its time complexity reached 286.24 million. For this kernel number the accuracy was reduced compared to the previous highest accuracy. We then changed the kernel size to 32, resulting in an improved model performance, recording a test accuracy of 95.59% and a training time per epoch of 77 s. Afterwards, we changed the kernel number to 64, resulting in a 96.29% testing accuracy with an epoch time of 78 s. Though the time complexity and training time was longer for 64 kernels, there was a noticeable gap in accuracy between 32 and 64 kernels. Therefore, we continue with a kernel number of 64.

Ablation Study 4: Altering the type of pooling layer.

Two types of pooling layers, the max-pooling layer and the average pooling layer [52], are evaluated in this ablation study. Both pooling layers achieved the same accuracy of 96.29%. It is observed that the time per epoch (77 s) and the time complexity (294.4 million) was also the same for both configurations. The max-pooling layer is chosen for further ablation study.

Ablation Study 5: Altering activation functions.

Since different activation functions can affect the performance of a CNN model, selection of the best activation function is essential. In this study, five activation functions, namely, ReLU, PReLU, Leaky ReLU, exponential linear units (ELU), and hyperbolic tangent (Tanh) [53] were evaluated where PReLU achieved the highest accuracy (97.78%). Altering the activation function did not affect the training time and time complexity. Hence, PReLU activation is chosen for further experimentation.

Ablation Study 6: Altering the batch size.

The batch size indicates the number of images that are used in each epoch to train the model. Usually, for a comparatively larger batch size, models take a longer period to achieve convergence. On the other hand, a smaller batch size may decrease the performance. Furthermore, the performance of the model varies depending on the batch size. Therefore, we tested the proposed model with a total of four different batch sizes (16, 32, 64, and 128). While the complexity (294.4 million) remained the same for all batch sizes, the test accuracy was different. For a batch size of 32, the highest accuracy of 97.78% was achieved with a time per epoch of 77 s. Therefore, we choose a batch size of 32 for further ablation studies.

Ablation Study 7: Altering the flatten layer.

Generally, the flatten layer produces a one-dimensional tensor from the multi-dimensional output of the previous layers. Here, we experimented with three types of layers, flatten, global average pooling and global max-pooling, as these layers are often used to convert variable size images into a convoluted feature of fixed size embedding [54]. However, as can be seen from Table 4, the highest accuracy of 97.89% and the shortest training time of 77 s per epoch and 294.4 million time of complexity was achieved with the flatten layer.

Ablation Study 8: Altering the loss function.

In this study, we experimented with different loss functions: binary cross-entropy, Kullback Leibler divergence, categorial cross-entropy, mean square error, mean absolute error, and mean squared logarithmic error. Since categorial cross-entropy achieves 97.89% accuracy with 77 s per epoch training time, this loss function is selected for further study.

Ablation Study 9: Altering the Optimizer.

Five different optimizers, namely, Adam, Nadam, Adamax, stochastic gradient descent (SGD), and root mean square propagation (RMSprop), were evaluated to identify the best optimizer. The highest testing accuracy was recorded with the Adam optimizer achieving a 98.03% accuracy and a time per epoch of 77 s with time complexity of 294.4 million. Thus, we choose the Adam optimizer for the final ablation study.

Ablation Study 10: Altering the learning rate.

We experimented with learning rates of 0.01, 0.001, 0.007, 0.0007, and 0.0001, where the highest accuracy of 98.34% was achieved with a learning rate of 0.0007. Therefore, we select the Adam optimizer with a learning rate of 0.0007.

After performing ten ablation studies, the optimal configuration of the proposed model architecture is summarized in Table 5.

### 5.4. Results and Analysis of Optimal Model

#### 5.4.1. Performance Evaluation Matrix of the Optimal Model

After completing the ablation studies of ten different cases on our base DCNN model, the classification accuracy improved from 95.59% to 98.34% by employing the optimal filter size, number of filters, pooling layers, activation function, batch size, flatten layer, loss function, optimizer, and learning rate. In this section, performance evaluation matrix, including sensitivity, specificity, precision, accuracy, F1-score, FPR, FNR, and FDR, are used to evaluate the model’s performance. Table 6 presents that our best-proposed model (InRes-106) achieved a sensitivity of 96.91%, a specificity of 98.01%, a precision of 97.74%, an accuracy of 98.34%, an F1-score of 96.14%, an FPR of 0.98%, an FNR of 5.74%, and an FDR of 2.26%. Moreover, after comparing the deep learning models’ (see Table 2) and InRes-106 models’ performance values, it can be concluded that all the performance values of our proposed model are quite satisfactory.

#### 5.4.2. Statistical Analysis of the Optimal Model

Table 7 represents the statistical analysis of the proposed model for the preprocessed dataset. The statistical values MAE, RMSE, SD, KC, MCC, [25] and a Wilcoxon signed-rank test are also considered. Cohen proposed the following interpretation of the Kappa value: if the KC value is less than or equal to 0, it indicates no agreement, 0.01 to 0.20 means none to slight, 0.21 to 0.40 is fair, 0.41 to 0.60 is moderate, 0.61 to 0.80 is substantial, and 0.81 to 1.00 is close to perfect agreement [55]. Furthermore, a Wilcoxon signed-rank test is also performed to highlight the statistical significance between the results produced by the proposed model and the other models [56]. In this circumstance, a *p*-value of less than 0.005 is considered a significant level [57]. Therefore, the Wilcoxon signed-rank test is directed to measure the *p*-value by comparing our proposed model (InRes-106) with two deep learning models: InceptionV3 and ResNet50. However, our proposed model achieved an MAE of 2.89%, RMSE of 11.62%, SD of 0.956, and KC of 97.604%. Since the KC value is 97.604%, our proposed model is closer to a perfect agreement. Additionally, the P value for InRes-106 vs. InceptionV3, and InRes-106 vs. ResNet50 are 0.004, and 0.004, respectively. Therefore, the results of this Wilcoxon signed-rank test demonstrate that the *p*-value for both cases is less than 0.005. Additionally, it is concluded that the performance difference between our proposed model, InRes-106, and other deep learning traditional models is statistically significant. Moreover, other values also indicate that our proposed model performs very well.

#### 5.4.3. Performance Analysis of the Optimal Model

The training and validity accuracy curve, the training and validity loss curve, the ROC curve, and our best model confusion matrix are also presented in this section.

Figure 12A represents the learning curve and the accuracy and the loss curve for our best-performing model. It can be seen that the training curve is seamlessly integrated from the first to the last epoch, almost without any hindrance. The gap between the validation accuracy curve and the training accuracy curve does not indicate any occurrence of overfitting during training. Similar to the training curve, the loss curve, shown in Figure 12B, converges steadily. Based on the training and the loss curve, it can be concluded that there is no evidence of overfitting issues. Figure 12C presents the confusion matrix of the best model. The row values represent the actual values of the test dataset, and the column values represent the predicted values of the test dataset. The diagonal value represents the TP values. It is found that the model is not biased towards any class and predicts all the five diseases almost equally. The ROC probability curve is plotted, and the area under the ROC Curve (AUC) value is taken from the ROC curve. The AUC value is a summary of the ROC curve representing the model’s ability to differentiate between various classes. If the AUC value is close to 1, the model is capable of detecting most classes. As shown in Figure 12D, the ROC curve is very close to touching the vertex of the y-axis, which means the true positive value is close to 1, and the false positive value is close to 0. In this study, we achieve an AUC value of 98.37%, which demonstrates the effectiveness of the proposed model.

### 5.5. K-Fold Cross Validation

K-fold cross-validation is a validation test that is executed using the training and test dataset [58]. First, the dataset is divided into k number of folds. Afterward, during the training and validation of k iterations, each have a separate fold of data for validation and training [59]. This method is used to observe the impact of randomness, bias, and variability, where the bias presents a difference between the actual accuracy and estimated accuracy [60]. The k-Fold cross-validation method is performed to evaluate the model’s robustness, stability and reliability. We experimented with 1-fold, 3-fold, 5-fold, 7-fold, and 9-fold cross-validations operations and achieved 98.22%, 98.27%, 98.29%, 98.34%, and 98.31% testing accuracy, respectively. Our best model achieved 98.34% as the highest testing accuracy. Based on the results, it can be concluded that all accuracies were close to the best-proposed model’s accuracy and that the performance did not drop drastically for any fold. Therefore, we can be assured that our proposed model will be able to achieve high accuracy even in a different training scenario with this dataset. Figure 13 presents the testing accuracy of different k-folds in the cross-validation method.

## 6. Discussion

Table 8 shows an overview of the comparison between relatable existing research studies and our proposed model.

In this section, we compare the results of our proposed InRes-106 model with some recent studies that were previously mentioned in the literature review section and used the same COVID and cardiac disease datasets with the same number of disease classes as their studies. Ozdemir et al. [15] introduced two binary classifications and one multi-class classification problem in their study where a highest accuracy of 98.57% was achieved for binary classification of COVID-19 vs. NHB. However, for multi-class classification, the model recorded quite a poor performance, achieving accuracies of 83.05% and 86.55%, respectively. In a fairly recent study conducted by Irmak et el. [16], the highest accuracy of 98.57% was recorded in binary classification, but during multi-classification of COVID-19 vs. AHB vs. MI, their model also achieved a poorer accuracy of 86.55%. Other researchers [17] using the same dataset found that after augmentation, the model accuracy dropped from 81.8% to 76.40%. Thus, looking at related previous work, it can be concluded that multi-class classification is more challenging than binary classification.

## 7. Conclusions

In this research, a multi-class classification problem is solved with high accuracy. A DCNN architecture is introduced, and the issue of time complexity and training time is taken into account. We have applied multiple image pre-processing techniques to the dataset to remove artifacts and enhance the image quality. After image pre-processing, five transfer learning models are trained and InceptionV3 and ResNet50 recorded the highest accuracy. Therefore, the proposed DCNN InRes-106 architecture is developed by integrating InceptionV3 and ResNet50, where ten different ablation case studies are performed to find the best configuration based on optimal performance. Initially, it was noted that over the 155–160 epochs, both the transfer learning models and the base model achieved their optimal accuracy. After completing the ablation study and using the best configuration, however, the proposed model achieved the highest testing accuracy in the 83rd epoch out of 160 epochs. This helps to reduce total training time. The proposed model outperforms the approaches of previous studies, achieving the highest accuracy of 98.34% with a low time complexity, due to the use of different image pre-processing techniques and the ablation study. Based on this research, it can be concluded that for the multi-class classification of CVD diseases using paper-based ECG reports, an improved performance can be achieved if more attention is given to the image pre-processing and model building techniques, including ablation study. Moreover, as ECG image datasets contain complex characteristics, DCNN architecture can be a suitable approach as it is able to extract and interpret meaningful and hidden features, resulting in high accuracy.

## 8. Limitations and Future Scope

The results indicate that the proposed approach is substantially better than various other classifiers that were achieved before in previous studies for multi-class classification of paper-based ECGs. However, this approach also has some limitations that can be addressed in the future, such as the fact that the dataset was highly imbalanced, consisting of only 1932 images; therefore, the number of images could be enhanced in future work. Another limitation is that there may be variability in the number of leads and derivations during ECG data acquisition. Furthermore, it could be explored how our proposed model will perform on real data. However, in most test cases, our best-proposed model performs very well, correctly classifying the five ECG classes. Despite minor limitations, this proposed model is very robust.

## Figures and Tables

**Figure 1 biomedicines-10-02835-f001:**
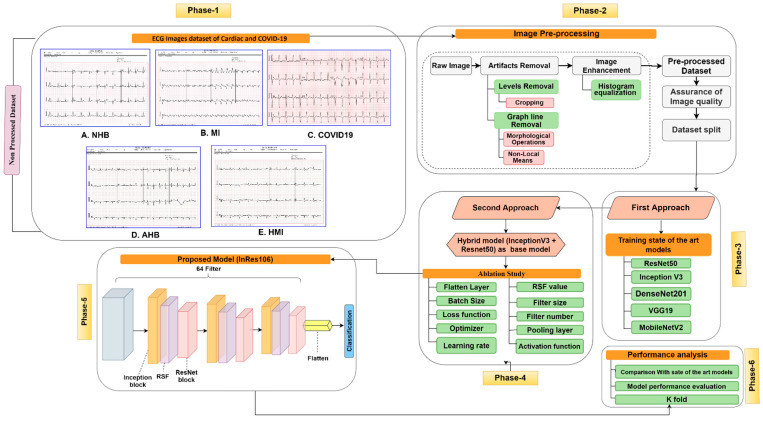
Flow diagram; Phase-1: Cardiac and COVID-19 image dataset, Phase-2: Image pre-processing (Raw image, Artifacts removal (labels removal, graph line removal), and image enhancement (HE), assurance of image quality, dataset split), Phase-3: First approach: Applying five transfer learning models (InceptionV3, ResNet50, MobileNetV2, VGG19, and DenseNet201), Phase-4: Second approach: Building a hybrid model combining InceptionV3 and ResNet50 model (base model), and applying ten ablation studies to the base model, Phase-5: Obtaining the best proposed model (InRes-106) after performing ablation studies, and Phase-6: Performance analysis (Comparison with sate of the art models, Model performance evaluation, and kfold cross-validation performance).

**Figure 2 biomedicines-10-02835-f002:**
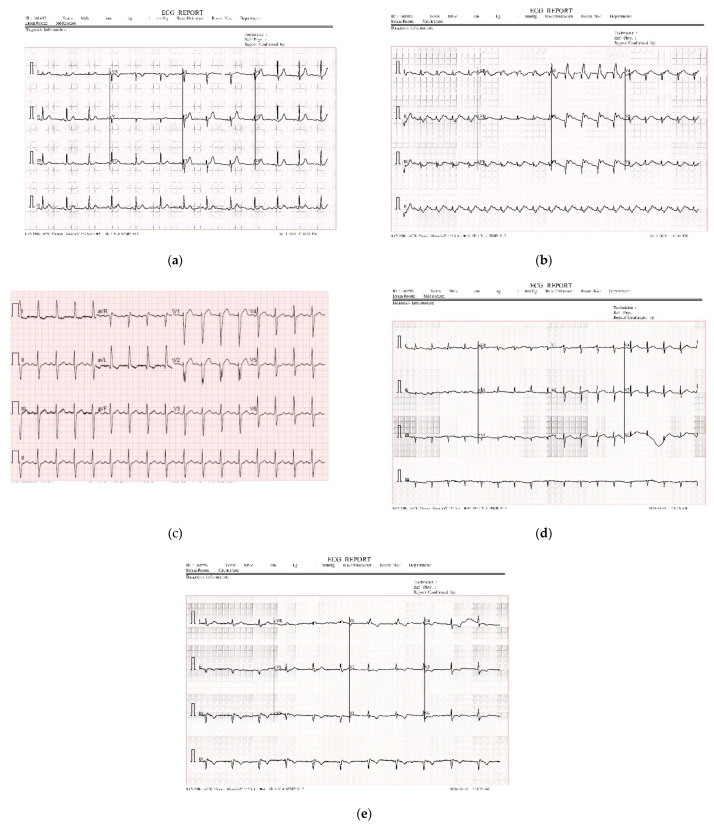
COVID and Cardiac disease dataset containing paper-based ECG signal images of five classes (**a**) NHB (**b**) MI (**c**) COVID-19 (**d**) AHB, and (**e**) HMI.

**Figure 3 biomedicines-10-02835-f003:**
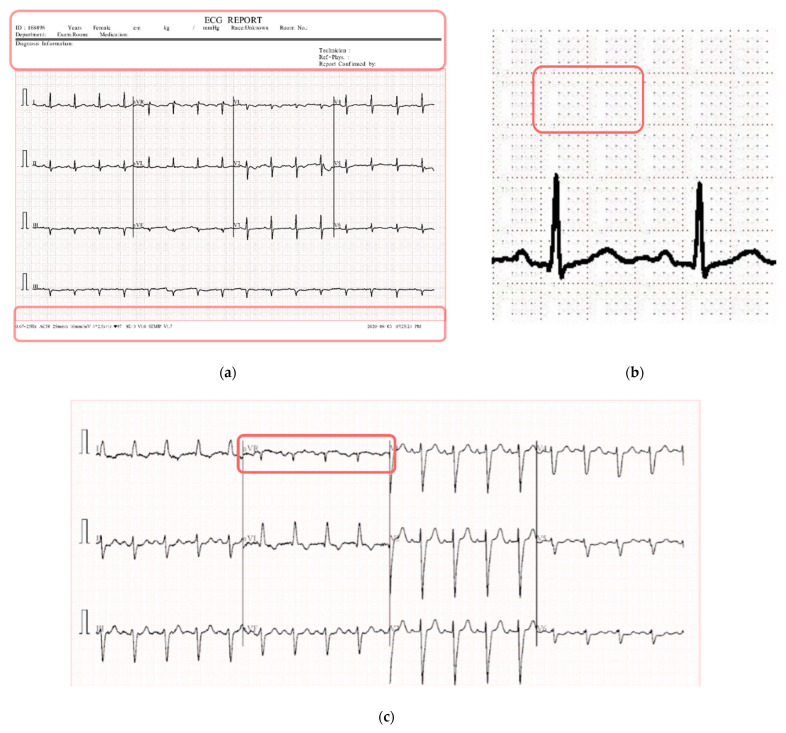
Challenges of the paper-based ECG image dataset (**a**) Different artifacts, (**b**) Background lines, and (**c**) Low contrast images are present in most classes.

**Figure 4 biomedicines-10-02835-f004:**
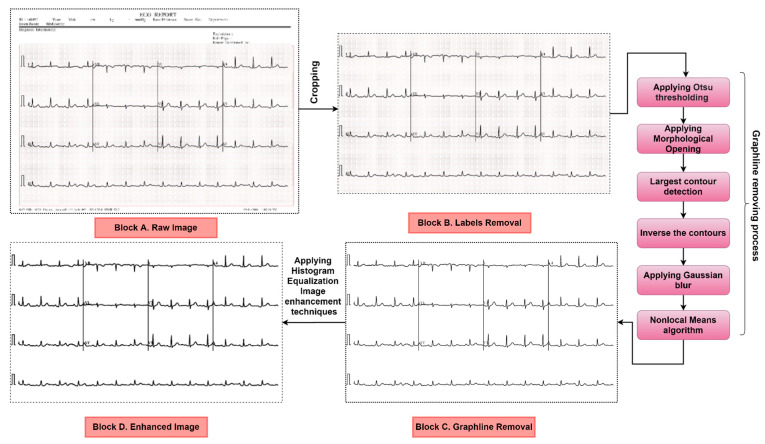
Image pre-processing techniques; Block (**A**–**C**) for artifact removal; Block (**A**): Raw image, Block (**B**): Label removal (by applying cropping techniques to the raw images to remove unusual texts and objects), Block (**C**): Graph line removal (applying Otsu thresholding, applying morphological opening, largest contour detection, inverse contour, Gaussian blur, and NLM), and Block (**D**): Applying HE technique to the pre-processed dataset to enhance image quality.

**Figure 5 biomedicines-10-02835-f005:**
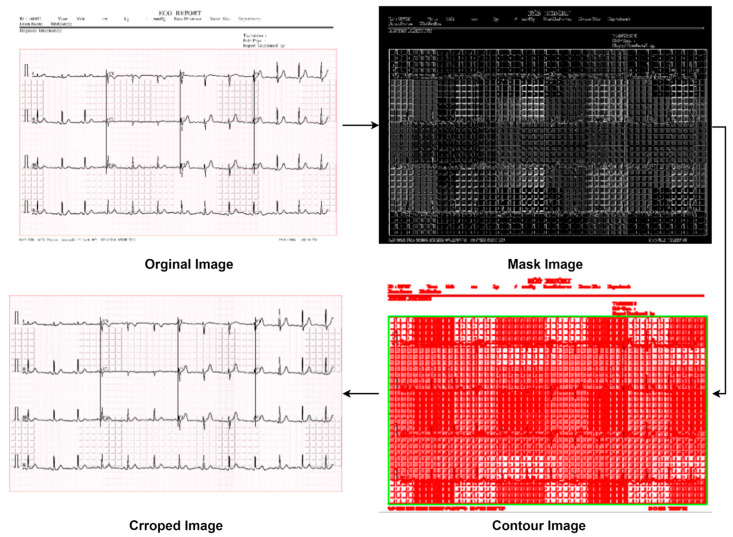
Label removal method.

**Figure 6 biomedicines-10-02835-f006:**
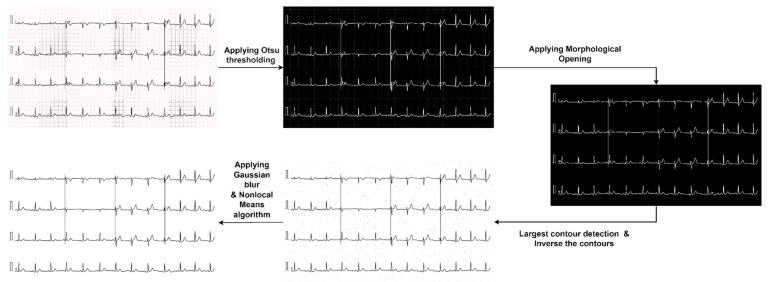
Graph line removal method.

**Figure 7 biomedicines-10-02835-f007:**
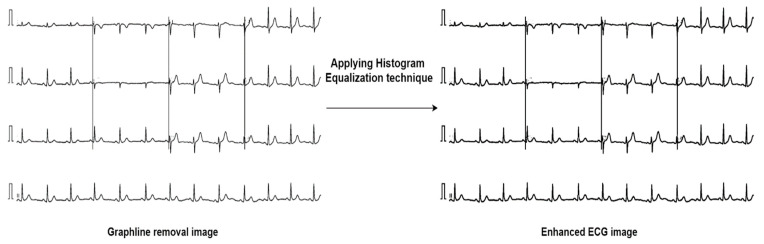
Enhance image quality using HE techniques.

**Figure 8 biomedicines-10-02835-f008:**
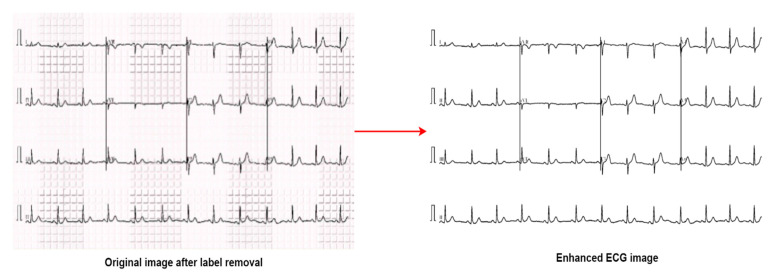
Original image (after label removal) and the enhanced image.

**Figure 9 biomedicines-10-02835-f009:**
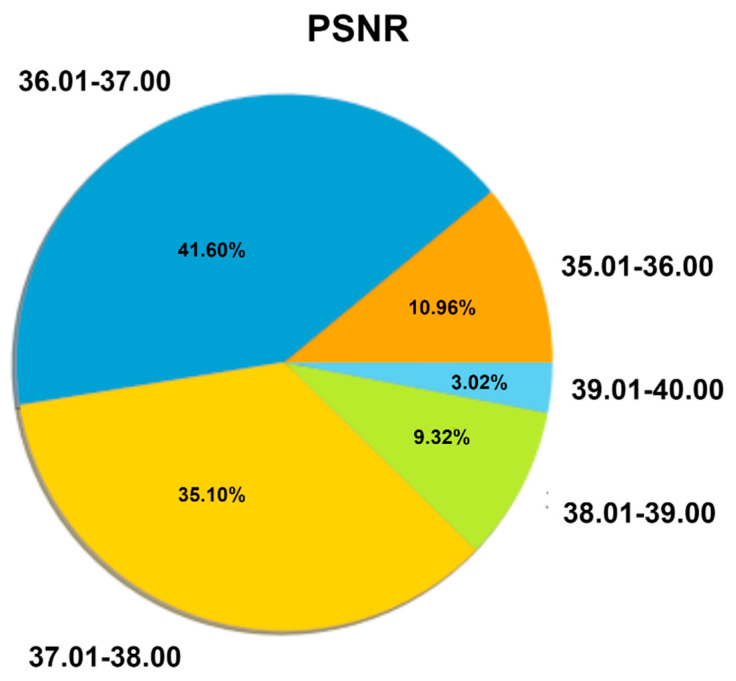
Pie chart of PSNR value for all 1932 ECG images.

**Figure 10 biomedicines-10-02835-f010:**
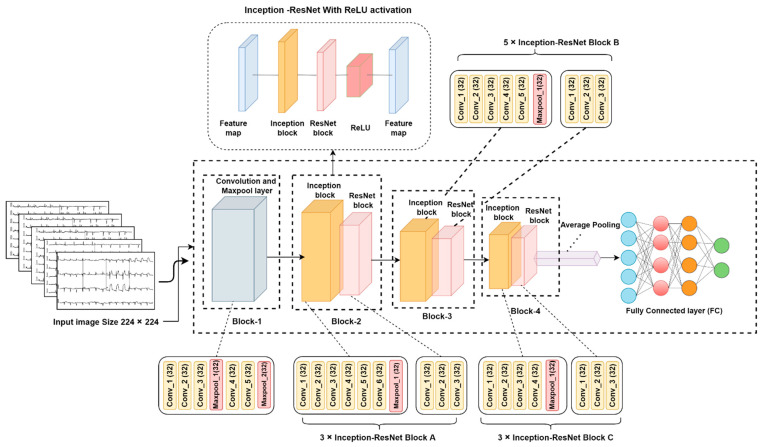
Base model architecture.

**Figure 11 biomedicines-10-02835-f011:**
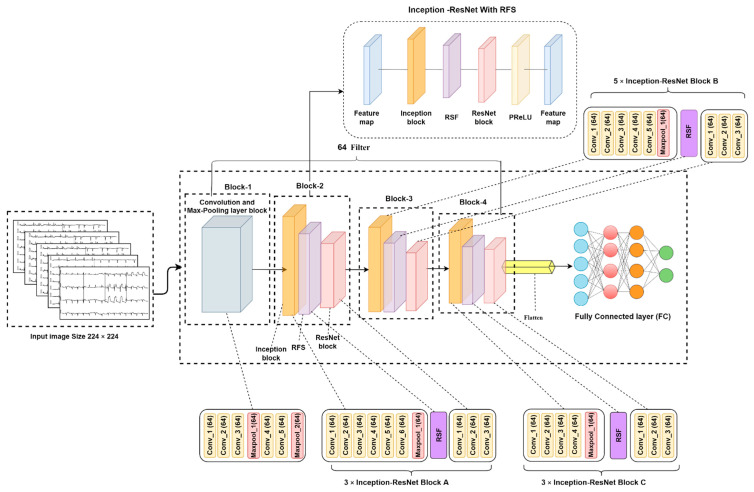
Proposed model architecture.

**Figure 12 biomedicines-10-02835-f012:**
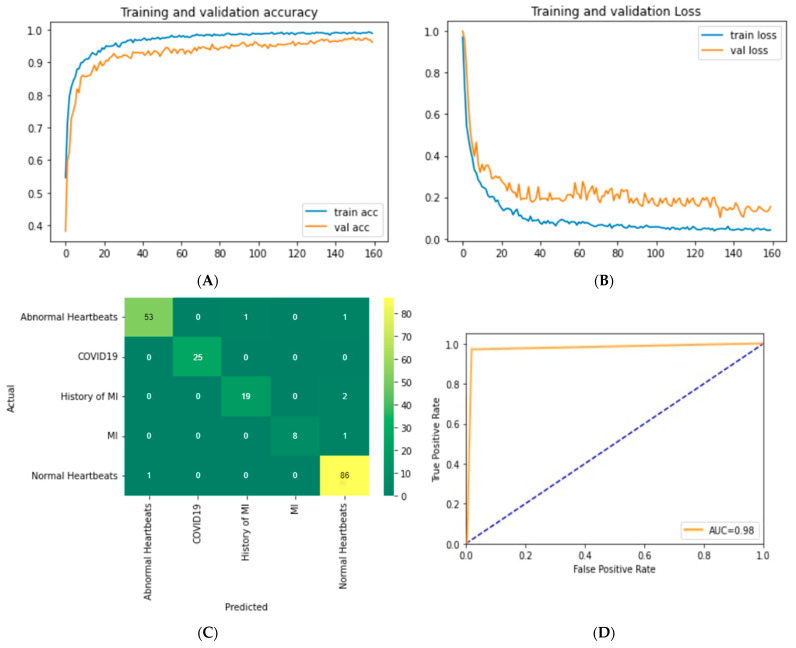
(**A**) Accuracy and (**B**) loss curve for training and validation over 160 epochs for Adam optimizer with a learning rate of 0.0007, (**C**) Confusion matrix (**D**) ROC curve for the optimal result of the proposed InRes-106 model after ablation study.

**Figure 13 biomedicines-10-02835-f013:**
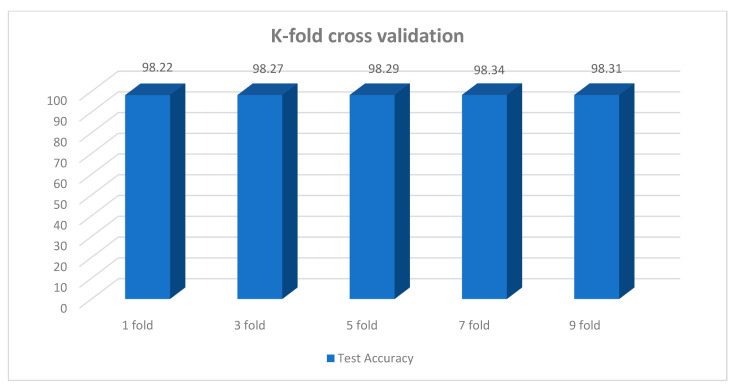
K-fold cross validation results.

**Table 1 biomedicines-10-02835-t001:** MSE, PSNR, SSIM and RMSE for twenty images.

Image	MSE	PSNR	SSIM	RMSE
Image_1	0.15	37.67	0.943	0.13
Image_2	0.14	38.57	0. 937	0.12
Image_3	0.14	37.58	0.944	0.12
Image_4	0.16	36.12	0.951	0.14
Image_5	0.13	36.68	0.948	0.11
Image_6	0.15	38.10	0.946	0.13
Image_7	0.16	37.33	0.942	0.14
Image_8	0.15	36.36	0.931	0.13
Image_9	0.16	37.14	0.937	0.13
Image_10	0.13	37.42	0.952	0.11
Image_11	0.14	36.14	0.946	0.13
Image_12	0.16	38.34	0.951	0.12
Image_13	0.12	35.69	0.943	0.12
Image_14	0.11	35.06	0.938	0.14
Image_15	0.15	37.11	0.952	0.11
Image_16	0.17	38.71	0.957	0.13
Image_17	0.14	36.78	0.939	0.14
Image_18	0.13	35.26	0.931	0.13
Image_19	0.16	37.45	0.932	0.13
Image_20	0.15	36.89	0.954	0.11

**Table 2 biomedicines-10-02835-t002:** Performance result analysis of the deep learning models on the pre-processed dataset.

Model Name	Specificity	Precision	Recall	F1-Score	Validation Accuracy	Validation Loss	Testing Accuracy	Testing Loss
InceptionV3	90.68	90.86	90.18	90.54	88.45	0.5128	90.56	0.3209
ResNet50	89.19	89.7	89.31	89.32	87.89	0.4434	89.63	0.3507
DenseNet201	93.59	85.23	81.67	82.90	86.78	0.5845	88.94	0.3802
VGG19	87.63	81.23	79.12	80.37	85.36	0.7233	87.87	0.3901
MobileNetV2	78.94	78.72	77.89	78.23	79.34	0.6216	80.56	0.4328

**Table 3 biomedicines-10-02835-t003:** Ablation study regarding layer configurations and activation functions.

**Ablation Study 1: Altering Residual Scaling Factor**					
**Configuration No.**	**RSF**	**Time Complexity**	**Epoch × Training Time**	**Test Accuracy (%)**	**Finding**
1	With RSF	434.5 M	160 × 77 s	**95.59**	Accuracy improved
2	Without RSF	657.3 M	160 × 84 s	95.01	Initial accuracy
**Ablation study 2: Altering the filter size**					
**Configuration No.**	**Filter size**	**Time complexity**	**Epoch × training time**	**Test accuracy (%)**	**Finding**
1	3 x 3	434.5 M	160 × 77 s	**95.59**	Identical accuracy
2	2 x 2	327.4 M	160 × 71 s	95.04	Accuracy dropped
3	5 x 5	717.4 M	160 × 85 s	95.894	Accuracy improved
**Ablation study 3: Altering the number of filters**					
**Configuration No.**	**No. of kernel**	**Time complexity**	**Epoch × training time**	**Test accuracy (%)**	**Finding**
1	16	286.24 M	160 × 74 s	95.23	Accuracy dropped
2	32	290.4 M	160 × 77 s	95.59	Identical Accuracy
3	64	294.4 M	160 × 78 s	**96.29**	Accuracy improved
**Ablation study 4: Altering the type of pooling layer**					
**Configuration No.**	**pooling layer**	**Time complexity**	**Epoch × training time**	**Test accuracy (%)**	**Finding**
1	Max	294.4 M	160 × 77 s	**96.29**	Identical accuracy
2	Average	294.4 M	160 × 77 s	96.29	Identical accuracy
**Ablation study 5: Altering the activation function**					
**Configuration No.**	**Activation**	**Time complexity**	**Epoch × training time**	**Test accuracy (%)**	**Finding**
1	PReLU	294.4 M	160 × 77 s	**97.28**	Accuracy improved
2	Relu	294.4 M	160 × 77 s	96.29	Identical accuracy
3	Leaky ReLu	294.4 M	160 × 77 s	95.55	Accuracy dropped
4	Tanh	294.4 M	160 × 77 s	96.11	Accuracy dropped
5	ELU	294.4 M	160 × 77 s	96.29	Identical accuracy

**Table 4 biomedicines-10-02835-t004:** Ablation study regarding model hyperparameters, loss function, and flatten layers.

**Ablation Study 6: Altering the Batch Size**					
**Configuration No.**	**Batch Size**	**Time Complexity**	**Epoch × Training Time**	**Test Accuracy (%)**	**Finding**
1	16	294.4 M	160 × 77 s	97.53	Accuracy dropped
2	32	294.4 M	160 × 77 s	**97.78**	Accuracy improved
3	64	294.4 M	160 × 78 s	97.28	Accuracy dropped
4	128	294.4 M	160 × 78 s	96.78	Accuracy dropped
**Ablation study 7: Altering the flatten layer**					
**Configuration No.**	**Flatten layer type**	**Time complexity**	**Epoch × training time**	**Test accuracy (%)**	**Finding**
1	Flatten	294.4 M	160 × 77 s	**97.89**	Accuracy improved
2	Global max-pooling	294.4 M	160 × 78 s	97.02	Accuracy dropped
3	Global average pooling	294.4 M	160 × 77 s	97.35	Accuracy dropped
**Ablation study 8: Altering the Loss Functions**					
**Configuration No.**	**Loss Function**	**Time complexity**	**Epoch × training time**	**Test accuracy (%)**	**Finding**
1	Binary cross-entropy	294.4 M	160 × 78 s	96.37	Accuracy dropped
2	Categorial cross-entropy	294.4 M	160 × 77 s	**97.89**	Identical accuracy
3	Mean squared error	294.4 M	160 × 77 s	96.73	Accuracy dropped
4	Mean absolute error	294.4 M	160 × 78 s	97.37	Accuracy dropped
5	Mean squared logarithmic error	294.4 M	160 × 78 s	96.81	Accuracy dropped
6	Kullback Leibler divergence	294.4 M	160 × 78 s	97.78	Accuracy dropped
**Ablation study 9: Altering the Optimizer**					
**Configuration No.**	**Optimizer**	**Time complexity**	**Epoch × training time**	**Test accuracy (%)**	**Finding**
1	Adam	294.4 M	160 × 77 s	**98.03**	Accuracy improved
2	Nadam	294.4 M	160 × 77 s	96.94	Accuracy dropped
3	SGD	297.5 M	160 × 78 s	96.58	Accuracy dropped
4	Adamax	294.4 M	160 × 77 s	97.89	Identical accuracy
5	RMSprop	294.4 M	160 × 79 s	96.08	Accuracy dropped
**Ablation study 10: Altering the Learning rate**					
**Configuration No.**	**Learning rate**	**Time complexity**	**Epoch × training time**	**Test accuracy (%)**	**Finding**
1	0.01	294.4 M	160 × 77 s	96.67	Accuracy dropped
2	0.007	294.4 M	160 × 77 s	97.46	Accuracy dropped
3	0.001	294.4 M	160 × 77 s	98.03	Identical accuracy
4	0.0007	294.4 M	160 × 78 s	**98.34**	Accuracy improved
5	0.0001	294.4 M	160 × 78 s	97.83	Accuracy dropped

**Table 5 biomedicines-10-02835-t005:** Configuration of the proposed architecture after ablation study.

Configuration	Value
Image size	224 × 224
Epochs	160
Optimization function	Adam
Learning rate	0.0007
Batch size	32
Number of filters	64
Filter size	3 × 3
Activation function	PReLU
Type of pooling layer	Max-pooling layer
Types of flatten layer	Flatten
Loss function	Categorial cross-entropy

**Table 6 biomedicines-10-02835-t006:** Performance evaluation matrix of the optimal configuration of proposed model.

Performance Analysis of the Best Configuration							
Sensitivity	Specificity	Precision	Accuracy	F1-Score	FPR	FNR	FDR
96.91	98.01	97.74	98.34	96.14	0.98	5.74	2.26

**Table 7 biomedicines-10-02835-t007:** Statistical result analysis of the optimal configuration of the proposed model.

Statistical Result Analysis of the Best Proposed Model				
MAE	RMSE	SD	KC	MCC
2.89	11.62	0.956	97.604	95.40

**Table 8 biomedicines-10-02835-t008:** Accuracy comparison with existing studies using the “ECG Images dataset of Cardiac and COVID-19”.

Reference	Model	Class Number	Batch size, Optimizer, and Learning Rate	Classification Process (Binary Class and Multi-Class Classification)	Accuracy
Ozdemir et al. (2021) [15]	Modified Alexnet (1 more convolutional layer + 256 filters + 3 × 3 kernel size)	4	128 Adam 0.0001	**Binary class:** COVID-19 and NHB	96.20%
				**Binary class:** Negative (NHB, AHB, and MI) and Positive (COVID-19)	93.00%
Irmak (2022) [16]	Proposed CNN model Inspired by VGG-16 model	4	32 Adam 0.0001	**Binary class:** COVID-19 vs. NHB	98.57%
				**Binary class:** COVID-19 vs. AHB	93.20%
				**Binary class:** COVID-19 vs. MI	96.74%
				**Multi-class:** COVID-19 vs. ABH vs. MI	86.55%
				**Multi- class:** COVID-19 vs. NBH vs. AHB vs. MI	83.05%
				**Binary class**: COVID-19 vs. NHB	98.57%
Anwar et al. (2021) [17]	EfficientNet B3	5	N/A Adam 0.0001	Before applying augmentation techniques	81.8%
				After applying augmentation techniques	76.40%
**Proposed work**	**InRes-106**	**5**	**64** **Adam** **0.0007**	**Multi-class: NHB vs. MI vs. COVID-19 vs. AHB vs. HMI**	**98.34%**

## Data Availability

The ECG Images dataset of Cardiac and COVID-19 Patients [13] used in this study is publicly available.

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
