# Peer review of "A Robust Framework Combining Image Processing and Deep Learning Hybrid Model to Classify Cardiovascular Diseases Using a Limited Number of Paper-Based Complex ECG Images"

_biomedicines, 2022, doi:10.3390/biomedicines10112835_

Round 1
Reviewer 1 Report
Please consider the following adjustments to increase clarity of the text:
- Use of simplified English;
- Do not use excessively long sentences (multiple compound / more than 5 lines).
Examples of possible changes:
Line 21-23: In the second approach, which is able to extract hidden and high-level features from images, an integrated deep learning model (InRes-106) is introduced, combining InceptionV3 and ResNet50 as a deep convolutional neural network (DCNN).
Line 43: According to a study conducted by British Heart Foundation, there are 7.6 million people in the UK with various cardiac diseases...
Line 84: Since several papers in the literature utilized a paper-based image dataset in their work, there were some major limitations or knowledge gaps. For example in several studies [15-18] researcher did ...
Another example is sentence 96-101, which can easily be divided into two separate sentences.
Line 125 Second bullet?
Line 130: Several accuracy matrices such as: sensitivity, specificity, precision, ...
Line 300: Then, only the contours with pixel size<50 are ...
Line 318: . In this process, the lost ...
Author Response
Response to Reviewer 1 Comments
Comments:
Examples of possible changes:
Comment 1: Line 21-23: In the second approach, which is able to extract hidden and high-level features from images, an integrated deep learning model (InRes-106) is introduced, combining InceptionV3 and ResNet50 as a deep convolutional neural network (DCNN).
Comment 2: Line 43: According to a study conducted by British Heart Foundation, there are 7.6 million people in the UK with various cardiac diseases...
Comment 3: Line 84: Since several papers in the literature utilized a paper-based image dataset in their work, there were some major limitations or knowledge gaps. For example, in several studies [15-18] researcher did ...
Comment 4: Another example is sentence 96-101, which can easily be divided into two separate sentences.
Comment 5: Line 125 Second bullet?
Comment 6: Line 130: Several accuracy matrices such as: sensitivity, specificity, precision, ...
Comment 7: Line 300: Then, only the contours with pixel size<50 are ...
Comment 8: Line 318: In this process, the lost ...
|
Number of Comment |
Action |
Refers to |
|
1 |
Addressed |
Abstract, Page-1, line (20-25) |
|
2 |
Addressed |
1. Introduction, Page1-2, line (43-46) |
|
3 |
Addressed |
2. Research aims and major contribution of the paper, Page-2, line (81-92) |
|
4 |
Addressed |
2. Research aims and major contribution of the paper, Page2-3, line (94-98) |
|
5 |
Addressed |
Remove from “2. Research aims and major contribution of the paper” section, Page-3 |
|
6 |
Addressed |
2. Research aims and major contribution of the paper, Page-3, line (145-149) |
|
7 |
Addressed |
4.2.1.2 Graph line Removal, Page-12, line 318 |
|
8 |
Addressed |
4.4.4. Image Enhancement, Page-12, line 338. |
Reviewer 2 Report
In this paper, the authors tackled an important problem of classifying cardiovascular diseases from ECG images. The topic is certainly worthy of investigation, but the manuscript suffers from the following shortcomings which need to be thoroughly addressed before the paper could be considered for publication:
1. The abstract is unnecessarily packed with acronyms – I suggest moving them to the main body of the manuscript. Also, please make sure that each acronym is defined exactly once at its first use (as an example, CVD is defined multiple times, both in the abstract and in the main text).
2. What is “an intensive innovation”? Please rephrase. Also, I would not say that using image pre-processing and ablation studies to design a pipeline for ECG analysis is “intensively novel”.
3. Please revise the bullet points summarizing the potential novelties (lines 110-137). What is “Second bullet”? Additionally, some of those points are rather vague and unnecessary – please present only the valid novelties and contributions (the quality should be valued here, not quantity).
4. Please improve the quality of figures (they should be in a vector format), as some of them are hardly readable after zooming (e.g., Figure 1).
5. The authors mix conceptualization with implementation – please replace the actual opencv methods presented in Figure 4 with the pseudocode/high-level names of those methods. Also, please present the motivation behind designing this very pre-processing pipeline in detail.
6. Some pre-processing steps seem to be designed ad hoc for a specific dataset (e.g., the cropping procedure). How does it generalize over different datasets? Similarly, there seem to be lots of “magic numbers” here and there – the same question applies here.
7. In Section 4.2.3, please present example pairs of images for which the quality metrics are calculated. This will help better understand their actual meaning.
8. Please present an illustrative example for 2-3 selected ECG images visualizing all pre-processing steps and the intermediate results.
9. Please provide the detailed dataset split into training/validation/test to ensure reproducibility (i.e., the IDs of images which are training/validation/test images).
10. Table 7 is quite misleading, as the gathered methods were executed over different datasets and likely dataset splits. To perform fair comparison, all methods should be confronted over the very same dataset, following the very same dataset split and validation procedure.
11. Please back up the claims with appropriate statistical testing, in order to verify if the differences across the investigated algorithms are statistically significant.
12. The manuscript would certainly benefit from careful proofreading, as I spotted quite a number of grammar errors and typos (e.g., “of a various deep learning classifiers”).
Author Response
Response to Reviewer 2 Comments
Comments:
Comment 1: The abstract is unnecessarily packed with acronyms – I suggest moving them to the main body of the manuscript. Also, please make sure that each acronym is defined exactly once at its first use (as an example, CVD is defined multiple times, both in the abstract and in the main text).
Author response: First of all, thank you very much for the time that you have given for the thorough review of our paper. We highly appreciate that you have pointed out this important aspect of the proposition.
Author action: All the acronyms are eliminated from the abstract (Page: 1, Line: 14-15) and moved them to the main body of the manuscript.
Comment 2: What is “an intensive innovation”? Please rephrase. Also, I would not say that using image pre-processing and ablation studies to design a pipeline for ECG analysis is “intensively novel”.
Author response: Thank you for your concern.
Author action: “an intensive innovation” is rephrased in (Page: 3, Line: 110-115).
Comment 3: Please revise the bullet points summarizing the potential novelties (lines 110-137). What is “Second bullet”? Additionally, some of those points are rather vague and unnecessary – please present only the valid novelties and contributions (the quality should be valued here, not quantity).
Author response: Thank you for your useful concern regarding the potential novelties. We have addressed your concern accordingly.
Author action: We have revised the bullet points and eliminated the “Second bullet” that is mistakenly stay in the bullet points. Additionally, only valid novelties and contributions are presented in the manuscript (Page: 3, Line: 116-151).
Comment 4: Please improve the quality of figures (they should be in a vector format), as some of them are hardly readable after zooming (e.g., Figure 1).
Author response: Thank you for pointing this out and we highly appreciate this feedback. Your observation in this regard is completely understandable.
Author action: We have replaced all figures with high-resolution images; hopefully, they should be readable now. Please check all images.
Comment 5: The authors mix conceptualization with implementation – please replace the actual opencv methods presented in Figure 4 with the pseudocode/high-level names of those methods. Also, please present the motivation behind designing this very pre-processing pipeline in detail.
Author response: Thank you for pointing this out. There is certainly some lacking in presenting in “Figure 4” which should have been pointed out.
Author action: According to your concern, we have replaced the actual opencv methods with the high-level names of those methods (please check figure 4) (Page: 10, Line:272-282). Additionally, the main motivation behind designing this very pre-processing pipeline is demonstrated in (Page: 9, Line: 267-271).
Comment 6: Some pre-processing steps seem to be designed ad hoc for a specific dataset (e.g., the cropping procedure). How does it generalize over different datasets? Similarly, there seem to be lots of “magic numbers” here and there – the same question applies here.
Author response: Thank you for your concern and pointing out this issue.
Author action: Proper explanation regarding this concern is added to the manuscript (Page: 11, Line: 297-302), and (Page: 12, Line: 308-311).
Comment 7: In Section 4.2.3, please present example pairs of images for which the quality metrics are calculated. This will help better understand their actual meaning.
Author response: Thank you for this observation and we have addressed this in the manuscript properly.
Author action: We have added a pair of images in section 4.2.3 and the proper explanation has been added in the manuscript in (Page: 13-14, Line: 369-377) for clarification.
Comment 8: Please present an illustrative example for 2-3 selected ECG images visualizing all pre-processing steps and the intermediate results.
Author response: This is a very impactful suggestion; the proper modifications had been made accordingly.
Author action: For visualizing all preprocessing steps and the intermediate results, three images including label removal (page-11, line 297-304), graph line removal (page-12, line 332-335), and enhanced images (page-11, line 356-357) are added in the manuscript.
Comment 9: Please provide the detailed dataset split into training/validation/test to ensure reproducibility (i.e., the IDs of images which are training/validation/test images).
Author response: Thank you for your concern.
Author action: Since the list Since the ECG image ID list is too long, we have uploaded the file to GitHub. Please go through the GitHub link to check (page-15, line 405-406).
Comment 10: Table 7 is quite misleading, as the gathered methods were executed over different datasets and likely dataset splits. To perform fair comparison, all methods should be confronted over the very same dataset, following the very same dataset split and validation procedure.
Author response: We fully appreciate your concern and have addressed them accordingly.
Author action: According your concern, we have sorted and organized the table 7. In addition, we have presented a comparison among those literature who used the same dataset that we used in our literature. So, please check (page-29-30, line 802-804).
Comment 11: Please back up the claims with appropriate statistical testing, in order to verify if the differences across the investigated algorithms are statistically significant.
Author response: A useful and valuable point of concern that needs more explanation and we have addressed it accordingly.
Author action: Proper explanation regarding this concern is added to the manuscript (Page: 25-28, Line: 719-780), including statistical testing of our model.
Comment 12: The manuscript would certainly benefit from careful proofreading, as I spotted quite a number of grammar errors and typos (e.g., “of a various deep learning classifiers”).
Author response: The stated concern regarding grammatical issues has been resolved.
Author action: The manuscript has been revised by a native speaker and all the grammatical errors are resolved.
Round 2
Reviewer 2 Report
Thank you indeed for addressing my concerns, well done!